# PoRF: Pose Residual Field for Accurate Neural Surface Reconstruction

**Jia-Wang Bian**
Department of Engineering Science
University of Oxford

**Wenjing Bian**
Department of Engineering Science
University of Oxford

**Victor Adrian Prisacariu**
Department of Engineering Science
University of Oxford

**Philip Torr**
Department of Engineering Science
University of Oxford

## Abstract

Neural surface reconstruction is sensitive to the camera pose noise, even if state-of-the-art pose estimators like COLMAP or ARKit are used. More importantly, existing Pose-NeRF joint optimisation methods have struggled to improve pose accuracy in challenging real-world scenarios. To overcome the challenges, we introduce the pose residual field (**PoRF**), a novel implicit representation that uses an MLP for regressing pose updates. This is more robust than the conventional pose parameter optimisation due to parameter sharing that leverages global information over the entire sequence. Furthermore, we propose an epipolar geometry loss to enhance the supervision that leverages the correspondences exported from COLMAP results without the extra computational overhead. Our method yields promising results. On the DTU dataset, we reduce the rotation error by 78% for COLMAP poses, leading to the decreased reconstruction Chamfer distance from 3.48mm to 0.85mm. On the MobileBrick dataset that contains casually captured unbounded 360-degree videos, our method refines ARKit poses and improves the reconstruction F1 score from 69.18 to 75.67, outperforming that with the dataset provided ground-truth pose (75.14). Moreover, we integrate our method into the Nerfstudio library, leading to consistently improved performance in diverse challenging scenes. These achievements demonstrate the efficacy of our approach in refining camera poses in real-world scenarios.

## 1 Introduction

Object reconstruction from multi-view images is a fundamental problem in computer vision. Recently, neural surface reconstruction (NSR) methods have significantly advanced in this field Wang et al. (2021a); Wu et al. (2023). These approaches draw inspiration from implicit scene representation and volume rendering techniques that were used in neural radiance fields (NeRF) Mildenhall et al. (2020). In NSR, scene geometry is represented by using a signed distance function (SDF) field, learned by a multilayer perceptron (MLP) network trained with image-based rendering loss. Despite achieving high-quality reconstructions, these methods are sensitive to camera pose noise, a common issue in real-world applications, even when state-of-the-art pose estimation methods like COLMAP Schönberger & Frahm (2016) or ARKit are used. An example of this sensitivity is evident in Fig. 1, where the reconstruction result with the COLMAP estimated poses shows poor quantitative accuracy and visible noise on the object's surface. In this paper, we focus on refining the inaccurate camera pose to improve neural surface reconstruction.

Recent research has explored the joint optimisation of camera pose and NeRF, while they are not designed for accurate neural surface reconstruction in real-world scenes. As shown in Fig. 1, existing methods such as BARF Lin et al. (2021) and SPARF Truong et al. (2023) are hard to improve reconstruction accuracy on the DTU dataset Jensen et al. (2014) via pose refinement. We regard the challenges as stemming from independent pose representation and weak supervision. Firstly, existing methods typically optimise pose parameters for each image independently. This approach

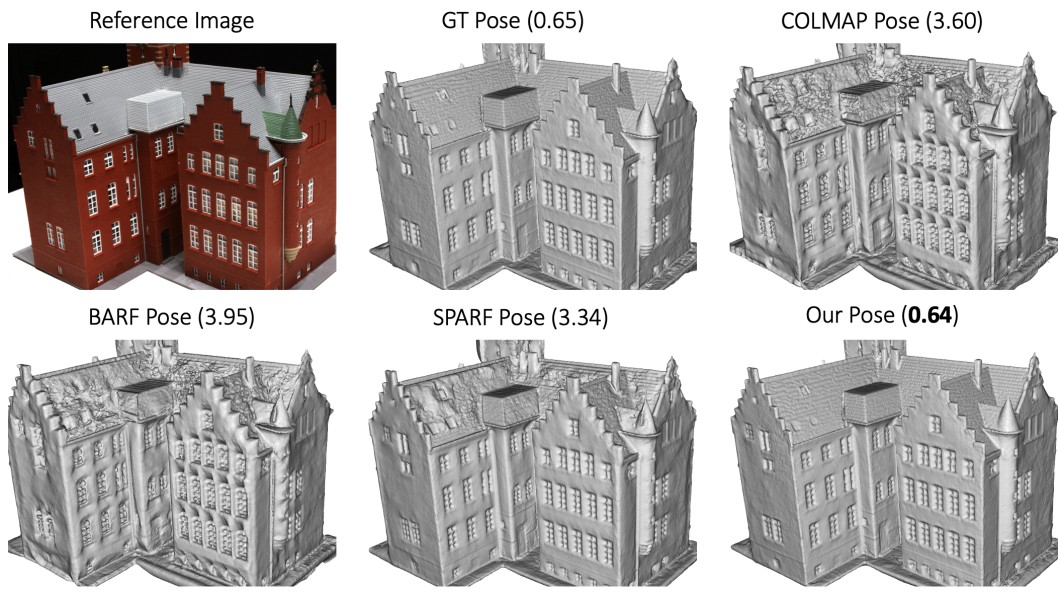

Figure 1: Reconstruction results on the DTU dataset (*scan24*). All meshes were generated by using Voxurf Wu et al. (2023). The Chamfer Distance (mm) is reported. BARF Lin et al. (2021), SPARF Truong et al. (2023), and our method all take the COLMAP pose as the initial pose. More results are illustrated in the supplementary material.

disregards global information over the entire sequence and leads to poor accuracy. Secondly, the colour rendering loss employed in the joint optimisation process exhibits ambiguity, creating many false local minimums. To overcome these challenges, we introduce a novel implicit pose representation and a robust epipolar geometry loss into the joint optimisation framework.

The proposed implicit pose representation is named *Pose Residual Field* (PoRF), which employs an MLP network to learn the pose residuals. The MLP network takes the frame index and initial camera pose as inputs, as illustrated in Fig. 2. Notably, as the MLP parameters are shared across all frames, it is able to capture the underlying global information over the entire trajectory for boosting performance. As shown in Fig. 3, the proposed PoRF shows significantly improved accuracy and faster convergence than the conventional pose parameter optimisation approach.

Moreover, we use feature correspondences in the proposed epipolar geometry loss to enhance supervision. Although correspondences have been used in SPARF Truong et al. (2023), it depends on expensive dense matching. As shown in Fig. 3, our method demonstrates similarly excellent performance with both sparse matches by handcraft methods (SIFT Lowe (2004)) and dense matches by deep learning methods (LoFTR Sun et al. (2021)). Besides, in contrast to SPARF which fuses correspondences and NeRF-rendered depths to compute reprojection loss, our approach uses the epipolar geometry loss that solely involves poses and correspondences, without relying on the rendered depths by NeRF that can be inaccurate and limit pose accuracy. Notably, as NeRF rendering is time-consuming, SPARF is constrained in the number of correspondences it can use in each iteration, while our method allows for the utilisation of an arbitrary number of matches.

We conduct a comprehensive evaluation on both DTU Jensen et al. (2014) and MobileBrick Li et al. (2023a) datasets. Firstly, on the DTU dataset, our method takes the COLMAP pose as initialisation and reduces the rotation error by 78%. This decreases the reconstruction Chamfer Distance from 3.48mm to 0.85mm, where we use Voxurf Wu et al. (2023) for reconstruction. Secondly, on the MobileBrick dataset, our method is initialised with ARKit poses and increases the reconstruction F1 score from 69.18 to 75.67. It slightly outperforms the provided imperfect GT pose in reconstruction (75.14) and achieves state-of-the-art performance. Moreover, we integrate our method into the Nerfstudio library Tancik et al. (2023), and the resulting method outperforms the original pose optimisation method in different challenging scenes. These results demonstrate the effectiveness of our approach in refining camera poses and improving reconstruction accuracy, especially in real-world scenarios where the initial poses might not be perfect.

In summary, our contributions are as follows:

1. We introduce a novel approach that optimises camera pose within neural surface reconstruction. It utilises the proposed *Pose Residual Field* (PoRF) and a robust epipolar geometry loss, enabling accurate and efficient refinement of camera poses.

2. The proposed method demonstrates its effectiveness in refining both COLMAP and ARKit poses, resulting in high-quality reconstructions on the DTU dataset and achieving state-of-the-art performance on the MobileBrick dataset.

## 2 RELATED WORK

**Neural surface reconstruction.** Object reconstruction from multi-view images is a fundamental problem in computer vision. Traditional multi-view stereo (MVS) methods Schönberger et al. (2016); Furukawa et al. (2009) explicitly find dense correspondences across images to compute depth maps, which are fused together to obtain the final point cloud. The correspondence search and depth estimation processes are significantly boosted by the deep learning based approaches Yao et al. (2018; 2019); Zhang et al. (2020a). Recently, Neural Radiance Field (NeRF) Mildenhall et al. (2020) has been proposed to implicitly reconstruct the scene geometry, and it allows for extracting object surfaces from the implicit representation. To improve performance, VolSDF Yariv et al. (2021) and NeuS Wang et al. (2021a) proposed to use an implicit SDF field for scene representation. MonoSDF Yu et al. (2022) and NeurIS Wang et al. (2022a) proposed to leverage the external monocular geometry prior. HF-NeuS Wang et al. (2022b) introduced an extra displacement network for learning surface details. NeuralWarp Darmon et al. (2022) used patch warping to guide surface optimisation. Geo-NeuS Fu et al. (2022) and RegSDF Zhang et al. (2022) proposed to leverage the sparse points generated by SfM. Neuralangelo Li et al. (2023b) proposed a coarse-to-fine method on the hash grids. Voxurf Wu et al. (2023) proposed a voxel-based representation that achieves high accuracy with efficient training.

**Joint NeRF and pose optimisation.** NeRFmm Wang et al. (2021b) demonstrates the possibility of jointly learning or refining camera parameters alongside the NeRF framework Mildenhall et al. (2020). BARF Lin et al. (2021) introduces a solution involving coarse-to-fine positional encoding to enhance the robustness of joint optimisation. In the case of SC-NeRF Jeong et al. (2021), both intrinsic and extrinsic camera parameters are proposed for refinement. In separate approaches, SiNeRF Xia et al. (2022) and GARF Chng et al. (2022) suggest employing distinct activation functions within NeRF networks to facilitate pose optimisation. Nope-NeRF Bian et al. (2023) employs an external monocular depth estimation model to assist in refining camera poses. L2G Chen et al. (2023) puts forth a local-to-global scheme, wherein the image pose is computed from learned multiple local poses. Notably similar to our approach is SPARF Truong et al. (2023), which also incorporates correspondences and evaluates its performance on the challenging DTU dataset Jensen et al. (2014). However, it is tailored to sparse-view scenarios and lacks the accuracy achieved by our method when an adequate number of images is available.

## 3 PRELIMINARY

**Volume rendering with SDF representation** We adhere to the NeuS Wang et al. (2021a) methodology to optimise the neural surface reconstruction from multi-view images. The approach involves representing the scene using an implicit SDF field, which is parameterised by an MLP. To render an image pixel, a ray originates from the camera centre $o$ and extends through the pixel along the viewing direction $v$, described as $\{p(t) = o + tv | t \geq 0\}$. By employing volume rendering (Max, 1995), the colour for the image pixel is computed by integrating along the ray using $N$ discrete sampled points $\{p_i = o + t_i v | i = 1, ..., N, t_i < t_{i+1}\}$:

$$\hat{C}(r) = \sum_{i=1}^{N} T_i \alpha_i c_i, \ \ T_i = \prod_{j=1}^{i-1}(1 - \alpha_j), \tag{1}$$

In this equation, $\alpha_i$ represents the opacity value, and $T_i$ denotes the accumulated transmittance. The primary difference between NeuS and NeRF Mildenhall et al. (2020) lies in how $\alpha_i$ is formulated. In NeuS, $\alpha_i$ is computed using the following expression:

$$\alpha_i = \max\left(\frac{\Phi_s(f(p(t_i))) - \Phi_s(f(p(t_{i+1})))}{\Phi_s(f(p(t_i)))}, 0\right),\tag{2}$$

In this context, $f(x)$ is the SDF function, and $\Phi_s(x) = (1 + e^{-sx})^{-1}$ denotes the Sigmoid function, where the parameter $s$ is automatically learned during the training process.

**Neural surface reconstruction loss.** In line with NeuS Wang et al. (2021a), our objective is to minimise the discrepancy between the rendered colours and the ground truth colours. To achieve this, we randomly select a batch of pixels and their corresponding rays in world space $P = \{C_k, o_k, v_k\}$ from an image in each iteration. Here, $C_k$ represents the pixel colour, $o_k$ is the camera centre, and $v_k$ denotes the viewing direction. We set the point sampling size as $n$ and the batch size as $m$.

The neural surface reconstruction loss function is defined as follows:

$$\mathcal{L}_{NSR} = \mathcal{L}_{colour} + \lambda\mathcal{L}_{reg}.\tag{3}$$

The colour loss $\mathcal{L}_{colour}$ is calculated as:

$$\mathcal{L}_{colour} = \frac{1}{m}\sum_k \left\|\hat{C}_k - C_k\right\|_1.\tag{4}$$

The term $\mathcal{L}_{reg}$ incorporates the Eikonal term Gropp et al. (2020) applied to the sampled points to regularise the learned SDF, which can be expressed as:

$$\mathcal{L}_{reg} = \frac{1}{nm}\sum_{k,i}(\|\nabla f(\hat{\mathbf{p}}_{k,i})\|_2 - 1)^2.\tag{5}$$

Here, $f$ is the learned SDF, and $\hat{\mathbf{p}}_{k,i}$ represents the sampled points for the $k$-th pixel in the $i$-th ray. $\lambda$ controls the influence of the regularisation term in the overall loss function.

**Joint pose optimisation with NSR.** Inspired by the previous work Wang et al. (2021b); Lin et al. (2021) that jointly optimises pose parameters with NeRF Mildenhall et al. (2020), we propose a naive joint pose optimisation with neural surface reconstruction and use Eqn. 3 as the loss function. This is denoted as the baseline method in Fig. 3, and in the next section, we present the proposed method for improving performance.

## 4 METHOD

**Pose residual field (PoRF).** We approach camera pose refinement as a regression problem using an MLP. In Fig. 2, we illustrate the process where the input to the MLP network is a 7D vector, consisting of a 1D image index ($i$) and a 6D initial pose ($r, t$). The output of the MLP is a 6D pose residual ($\Delta r, \Delta t$). This residual is combined with the initial pose to obtain the final refined pose ($r', t'$). In this context, rotation is represented by 3D axis angles.

To encourage small pose residual outputs at the beginning of the optimisation, we multiply the MLP output by a fixed small factor ($\alpha$). This ensures an effective initialisation (the refined pose is close to the initial pose), and the correct pose residuals can be gradually learned with optimisation. Formally, the composition of the initial pose and the learned pose residual is expressed as follows:

$$\begin{cases} \hat{r} = r + \alpha \cdot \Delta r \\ \hat{t} = t + \alpha \cdot \Delta t \end{cases}\tag{6}$$

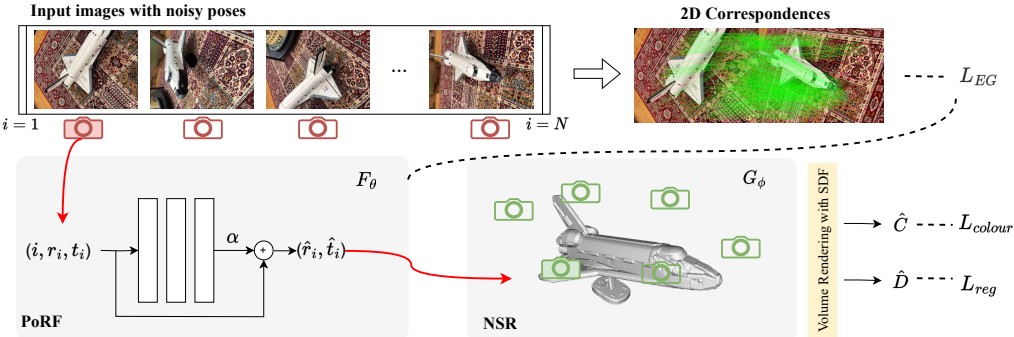

Figure 2: Joint optimisation pipeline. The proposed model consists of a pose residual field $F_\theta$ and a neural surface reconstruction module $G_\phi$. PoRF takes the frame index and the initial camera pose as input and employs an MLP to learn the pose residual, which is composited with the initial pose to obtain the predicted pose. The output pose is used to compute the neural rendering losses with the NSR module and the epipolar geometry loss with pre-computed 2D correspondences. Parameters $\theta$ and $\phi$ are updated during back-propagation.

The resulting final pose $(\hat{r}, \hat{t})$ is utilised to compute the loss, and the gradients can be backpropagated to the MLP parameters during the optimisation process. In practice, we use a shallow MLP with 2 hidden layers, which we find is sufficient for object reconstruction.

**Intuition on the PoRF.**   Our method draws inspiration from the coordinates-MLP. Similar to NeRF Mildenhall et al. (2020), in which the position and view direction of a point are taken as inputs by the MLP to learn its density and colour, our method takes the initial position (translation) and view direction (rotation) of the camera as inputs to learn the camera's pose residual. Also, our approach incorporates the frame index as an additional input, serving as a temporal coordinate.

The MLP parameters are shared across all frames, which enables our method to extract global information from the entire sequence. The inclusion of the frame index allows our method to learn local information for each frame. Overall, the utilisation of both global and local information empowers our method to effectively handle various challenges and learn accurate camera poses.

The parameter sharing in our method leads to better robustness than the conventional pose refinement method. Specifically, the previous method directly back-propagates gradients to per-frame pose parameters, so the noisy supervision and inappropriate learning rates can make certain pose parameters trapped in false local minima, causing the optimisation to diverge. In contrast, our method can leverage information from other frames to prevent certain frames from being trapped in false local minima. A quantitative comparison is illustrated in Fig. 3, where the performance of L2 (utilising PoRF) notably surpasses that of L1 (excluding PoRF).

**Epipolar geometry loss.**   To improve the supervision, we introduce an epipolar geometry loss that makes use of feature correspondences. The correspondences can be either sparse or dense, and they can be obtained through traditional handcrafted descriptors, like SIFT Lowe (2004), or deep learning techniques, like LoFTR Sun et al. (2021). As a default option, we utilise SIFT matches, exported from COLMAP pose estimation results, to avoid introducing extra computational cost.

In each training iteration, we randomly sample $n$ matched image pairs and calculate the Sampson distance (Eqn. 8) for the correspondences. By applying a threshold $\delta$, we can filter out outliers and obtain $m$ inlier correspondences. We represent the inlier rate for each pair as $p_i$, and the Sampson error for each inlier match as $e_k$. The proposed epipolar geometry loss is then defined as:

$$\mathcal{L}_{EG} = \frac{1}{n} \sum_i w_i \left( \frac{1}{m} \sum_k e_k \right). \tag{7}$$

Table 1: Absolute pose accuracy evaluation on the DTU dataset. For all methods, the COLMAP pose is used as the initial pose. The upper section presents rotation errors in degrees, while the lower section displays translation errors in millimetres (mm).

| Scan | 24 | 37 | 40 | 55 | 63 | 65 | 69 | 83 | 97 | 105 | 106 | 110 | 114 | 118 | 122 | mean |
|------|------|------|------|------|------|------|------|------|------|------|------|------|------|------|------|------|
| COLMAP (Initial) | 0.68 | 0.67 | 0.63 | 0.62 | 0.75 | 0.67 | 0.64 | 0.68 | 0.56 | 0.66 | 0.62 | 0.69 | 0.66 | 0.64 | 0.63 | 0.65 |
| BARF Lin et al. (2021) | 0.67 | 0.73 | 0.67 | 0.79 | 0.67 | 0.83 | 0.47 | 0.59 | 0.55 | 0.62 | 0.75 | 0.67 | 0.67 | 0.70 | 0.84 | 0.69 |
| L2G Chen et al. (2023) | 0.67 | 0.65 | 0.69 | 32.04 | 0.80 | 0.68 | 0.49 | 0.66 | 0.76 | 0.61 | 29.02 | 6.41 | 0.69 | 2.63 | 2.47 | 5.29 |
| SPARF Truong et al. (2023) | 0.40 | 0.62 | 0.43 | 0.40 | 0.72 | 0.39 | 0.30 | 0.31 | 0.48 | 0.33 | 0.36 | 0.39 | 0.19 | 0.24 | 0.36 | 0.39 |
| Ours | **0.09** | **0.13** | **0.10** | **0.13** | **0.13** | **0.16** | **0.13** | **0.12** | **0.26** | **0.12** | **0.12** | **0.16** | **0.15** | **0.11** | **0.13** | **0.14** |
| COLMAP (Initial) | 0.79 | **1.06** | 0.81 | 0.81 | 0.80 | 0.89 | 0.88 | **1.07** | 2.07 | 1.07 | 1.05 | 1.22 | 1.03 | **0.95** | **1.02** | 1.03 |
| BARF Lin et al. (2021) | 1.53 | 2.19 | 1.80 | 2.50 | 2.32 | 2.31 | 2.06 | 2.65 | 3.45 | 2.27 | 2.29 | 2.59 | 3.29 | 3.15 | 3.13 | 2.50 |
| L2G Chen et al. (2023) | 1.50 | 1.90 | 1.90 | 115 | 2.07 | 2.45 | 1.89 | 2.77 | 5.04 | 1.99 | 172 | 88.6 | 2.94 | 12.3 | 21.1 | 28.9 |
| SPARF Truong et al. (2023) | 3.66 | 4.76 | 3.89 | 2.22 | 5.68 | 2.57 | 2.32 | 2.96 | 4.88 | 3.21 | 2.94 | 2.85 | 1.96 | 2.23 | 2.65 | 3.25 |
| Ours | **0.76** | 1.11 | **0.77** | **0.80** | **0.79** | **0.81** | **0.84** | 1.08 | **2.05** | **1.02** | **1.03** | **1.15** | **0.99** | 0.97 | 1.07 | **1.02** |

Table 2: Reconstruction accuracy evaluation on the DTU dataset. The evaluation metric is Chamfer distance, expressed in millimetres (mm).

| Method | Pose | 24 | 37 | 40 | 55 | 63 | 65 | 69 | 83 | 97 | 105 | 106 | 110 | 114 | 118 | 122 | mean |
|--------|------|------|------|------|------|------|------|------|------|------|------|------|------|------|------|------|------|
| NeRF(Mildenhall et al., 2020) | | 1.83 | 2.39 | 1.79 | 0.66 | 1.79 | 1.44 | 1.50 | 1.20 | 1.96 | 1.27 | 1.44 | 2.61 | 1.04 | 1.13 | 0.99 | 1.54 |
| IDR(Yariv et al., 2020) | | 1.63 | 1.87 | 0.63 | 0.48 | 1.04 | 0.79 | 0.77 | 1.33 | 1.16 | 0.76 | 0.67 | 0.90 | 0.42 | 0.51 | 0.53 | 0.90 |
| VolSDFYariv et al. (2021) | GT | 1.14 | 1.26 | 0.81 | 0.49 | 1.25 | 0.70 | 0.72 | 1.29 | 1.18 | 0.70 | 0.66 | 1.08 | 0.42 | 0.61 | 0.55 | 0.86 |
| PointNeRF(Xu et al., 2022) | | 0.87 | 2.06 | 1.20 | 1.01 | 1.01 | 1.39 | 0.80 | **1.04** | **0.92** | 0.74 | 0.97 | **0.76** | 0.56 | 0.90 | 1.05 | 1.02 |
| NeuS(Wang et al., 2021a) | | 0.83 | 0.98 | 0.56 | 0.37 | 1.13 | **0.59** | **0.60** | 1.45 | 0.95 | 0.78 | **0.52** | 1.43 | **0.36** | **0.45** | **0.45** | 0.77 |
| Voxurf Wu et al. (2023) | | **0.65** | **0.74** | **0.39** | **0.35** | 0.96 | 0.64 | 0.85 | 1.58 | 1.01 | **0.68** | 0.60 | 1.11 | 0.37 | **0.45** | 0.47 | **0.72** |
| | COLMAP | 3.60 | 4.36 | 3.62 | 3.00 | 4.44 | 3.52 | 2.64 | 4.11 | 2.85 | 3.86 | 2.98 | 3.86 | 2.76 | 3.26 | 3.34 | 3.48 |
| Voxurf | BARF | 3.95 | 4.35 | 3.84 | 3.02 | 4.38 | 4.09 | 2.02 | 3.68 | 2.56 | 3.62 | 3.61 | 4.06 | 2.63 | 3.04 | 4.14 | 3.53 |
| | SPARF | 3.34 | 5.30 | 1.35 | 2.75 | 6.17 | 2.81 | 1.85 | 3.24 | 1.85 | 2.17 | 2.05 | 5.30 | 1.03 | 1.62 | 2.45 | 2.89 |
| | Ours | **0.64** | **0.84** | **0.59** | **0.48** | **1.10** | **1.02** | **0.90** | **1.43** | **1.10** | **0.83** | **0.86** | **1.11** | **0.58** | **0.65** | **0.67** | **0.85** |

Here, $w_i = p_i{}^2$ is the loss weight that mitigates the influence of poorly matched pairs. The computation of the Sampson distance between two points, denoted as $x$ and $x'$, with the fundamental matrix $F$ obtained from the (learned) camera poses and known intrinsics, is expressed as:

$$d_{\text{Sampson}}(x, x', F) = \frac{(x'^T F x)^2}{(Fx)_1^2 + (Fx)_2^2 + (F^T x')_1^2 + (F^T x')_2^2}. \tag{8}$$

Here, subscripts 1 and 2 refer to the first and second components of the respective vectors.

**Training objectives.** The overall loss function in our proposed joint optimisation method combines both Eqn. 3 and Eqn. 7, as follows:

$$\mathcal{L} = \mathcal{L}_{NSR} + \beta \mathcal{L}_{EG}, \tag{9}$$

where $\beta$ is a hyperparameter that controls loss weights. The first term $\mathcal{L}_{NSR}$ represents the loss related to neural surface reconstruction, which backpropagates the gradients to both PoRF and rendering network parameters. The second term $\mathcal{L}_{EG}$ represents the loss associated with the epipolar geometry loss, which backpropagates the gradients to PoRF only.

## 5  EXPERIMENTS

**Experiment setup.** We perform our experiments on both the DTU Jensen et al. (2014) and MobileBrick Li et al. (2023a) datasets. Following previous methods such as Wu et al. (2023) and Li et al. (2023a), our evaluation uses the provided 15 test scenes from DTU and 18 test scenes from MobileBrick. We assess both the accuracy of camera poses and the quality of surface reconstructions. The baseline methods for comparison includes BARF Lin et al. (2021), L2G Chen et al. (2023), and SPARF Truong et al. (2023). For a fair comparison, all methods, including ours, take the same initial pose inputs—namely, COLMAP poses on the DTU dataset and ARKit poses on the MobileBrick dataset. Here, we consider all methods to be operating in a two-stage manner, so only the pose refinement is conducted by these methods, and we utilise Voxurf Wu et al. (2023) with the refined poses for the reconstruction step for all methods.

Table 3: Absolute pose accuracy on the MobileBrick dataset. The ARKit pose serves as the initial pose for all methods. Note that the provided GT pose in this dataset is imperfect.

| Metrics | Rotation Error (Deg.) | | | | | Translation Error (mm) | | | | |
|---|---|---|---|---|---|---|---|---|---|---|
| Scene | ARKit | BARF | L2G | SPARF | Ours | ARKit | BARF | L2G | SPARF | Ours |
| aston | 0.43 | 0.22 | 0.22 | 0.22 | **0.14** | 2.49 | 1.21 | 1.21 | 1.01 | **0.70** |
| audi | 0.64 | 0.46 | 0.52 | 1.10 | **0.28** | 3.38 | 2.25 | 2.24 | 4.35 | **1.11** |
| beetles | 0.49 | 0.52 | 0.57 | 0.17 | **0.14** | 1.40 | 1.29 | 2.06 | **1.21** | 1.23 |
| big_ben | 0.44 | 0.33 | 0.49 | 0.27 | **0.26** | 4.23 | 3.03 | 2.13 | 2.38 | **1.82** |
| boat | 0.61 | 0.24 | 0.38 | 0.26 | **0.22** | 1.51 | 0.97 | 1.29 | 1.12 | **0.98** |
| bridge | 0.44 | 0.22 | 0.18 | 0.32 | **0.22** | 1.08 | 1.29 | 1.12 | 1.26 | **0.78** |
| cabin | 0.61 | 0.34 | 0.32 | 0.30 | **0.26** | 1.34 | 1.44 | 1.55 | 1.09 | **0.93** |
| camera | 0.30 | 0.23 | 0.26 | **0.17** | 0.18 | 0.86 | 1.06 | 1.07 | 0.67 | **0.61** |
| castle | 0.55 | 4.77 | 4.79 | 4.77 | **0.30** | 2.42 | 17.1 | 21.1 | 24.12 | **2.55** |
| colosseum | 0.46 | 0.49 | 0.35 | 0.95 | **0.28** | 3.67 | 4.21 | 4.41 | 11.26 | **3.25** |
| convertible | 0.40 | 0.27 | 0.25 | 0.23 | **0.13** | 1.65 | 1.18 | 1.21 | 1.31 | **0.92** |
| ferrari | 0.43 | 0.20 | 0.20 | 0.23 | **0.13** | 1.89 | 1.10 | 1.19 | 0.80 | **0.59** |
| jeep | 0.34 | 0.24 | 0.22 | 0.28 | **0.23** | 0.98 | 1.13 | 1.08 | 0.65 | **0.50** |
| london_bus | 0.72 | 0.70 | 0.73 | 0.50 | **0.42** | 3.62 | 2.97 | 2.98 | 3.61 | **2.63** |
| motorcycle | 0.34 | 0.26 | 0.29 | 0.27 | **0.20** | 0.97 | 1.01 | 1.10 | 0.82 | **0.60** |
| porsche | 0.28 | 0.19 | 0.20 | 0.20 | **0.15** | 1.29 | 1.16 | 1.18 | 0.88 | **0.71** |
| satellite | 0.41 | 0.23 | 0.26 | **0.15** | 0.17 | 1.45 | 0.92 | 1.11 | **0.97** | 0.98 |
| space_shuttle | 0.31 | 0.28 | 0.26 | 0.58 | **0.18** | 1.45 | 1.81 | 1.76 | 3.43 | **1.30** |
| mean | 0.46 | 0.57 | 0.58 | 0.61 | **0.22** | 1.90 | 2.50 | 2.84 | 3.39 | **1.23** |

Table 4: Reconstruction accuracy evaluation on the MobileBrick dataset. Note that the provided GT pose in this dataset is imperfect. The results are averaged over all 18 test scenes.

| Methods | Pose | $\sigma = 2.5mm$ | | | $\sigma = 5mm$ | | | Chamfer $\downarrow$ |
|---|---|---|---|---|---|---|---|---|
| | | Accu. (%) $\uparrow$ | Rec. (%) $\uparrow$ | F1 $\uparrow$ | Accu. (%) $\uparrow$ | Rec. (%) $\uparrow$ | F1 $\uparrow$ | (mm) |
| TSDF-Fusion Zhou et al. (2018) | | 42.07 | 22.21 | 28.77 | 73.46 | 42.75 | 53.39 | 13.78 |
| BNV-Fusion Li et al. (2022) | | 41.77 | 25.96 | 33.27 | 71.20 | 47.09 | 55.11 | 9.60 |
| Neural-RGBD Azinović et al. (2022) | | 20.61 | 10.66 | 13.67 | 39.62 | 22.06 | 27.66 | 22.78 |
| COLMAP Schönberger & Frahm (2016) | GT | 74.89 | 68.20 | 71.08 | 93.79 | 84.53 | 88.71 | 5.26 |
| NeRF Mildenhall et al. (2020) | | 47.11 | 40.86 | 43.55 | 78.07 | 69.93 | 73.45 | 7.98 |
| NeuS Wang et al. (2021a) | | 77.35 | 70.85 | 73.74 | 93.33 | 86.11 | 89.30 | 4.74 |
| Voxurf Wu et al. (2023) | | 78.44 | 72.41 | 75.13 | 93.95 | 87.53 | 90.41 | 4.71 |
| | ARKit | 71.90 | 66.98 | 69.18 | 91.81 | 85.71 | 88.43 | 5.30 |
| Voxurf | BARF | 77.12 | 71.36 | 73.96 | 93.47 | 87.56 | 90.22 | 4.83 |
| | SPARF | 76.10 | 70.44 | 72.99 | 92.73 | 86.66 | 89.39 | 4.95 |
| | Ours | **79.01** | **72.94** | **75.67** | **94.07** | **87.84** | **90.65** | **4.67** |

**Implementation details.** The hyperparameters used for training the surface reconstruction align with those utilised in NeuS Wang et al. (2021a). To calculate the proposed epipolar geometry loss, we randomly sample 20 image pairs in each iteration and distinguish inliers and outliers by using a threshold of 20 pixels. The entire framework undergoes training for 50,000 iterations, which takes 2.5 hours in one NVIDIA A40 GPU. However, it's noteworthy that convergence is achieved rapidly, typically within the first 5,000 iterations. More details are in the supplementary materials.

## 5.1 COMPARISONS

**Results on DTU.** The pose refinement results are presented in Table 1. It shows that BARF Lin et al. (2021) and L2G Chen et al. (2023) are unable to improve the COLMAP pose on DTU, and L2G diverges in several scenes. While SPARF Truong et al. (2023) shows a slight improvement in rotation, it remains limited, and it results in worse translation. Besides, we also evaluated Nope-NeRF Bian et al. (2023), but it diverged on DTU and MobileBrick scenes due to the bad initialisation of the depth scale. Overall, our proposed method achieves a substantial reduction of 78% in rotation error and shows a minor improvement in translation error.

The reconstruction results are presented in Tab. 2, where we run Voxurf Wu et al. (2023) for reconstruction. It shows that the reconstruction with the COLMAP pose falls behind previous methods that use the GT pose. BARF Lin et al. (2021) is hard to improve performance. While SPARF Truong et al. (2023) can slightly improve performance, it is limited. When employing the refined pose by our method, the reconstruction accuracy becomes comparable to most of the previous methods that

Table 5: Novel view rendering results on the Nerfstudio dataset. We replace the pose optimisation module in the Nerfacto with our PoRF MLP, leading to consistently improved novel view rendering accuracy. Note that we do not use the proposed loss function here.

| Scenes | Ours | | | Nerfacto | | |
|---|---|---|---|---|---|---|
| | PSNR ↑ | SSIM ↑ | LPIPS ↓ | PSNR | SSIM | LPIPS |
| Egypt | **23.08** | **0.743** | **0.245** | 20.75 | 0.623 | 0.305 |
| person | **25.87** | **0.742** | **0.261** | 24.33 | 0.643 | 0.299 |
| kitchen | **20.72** | **0.796** | **0.271** | 19.60 | 0.73 | 0.298 |
| plane | **22.44** | **0.672** | **0.356** | 19.72 | 0.557 | 0.440 |
| dozer | **21.16** | **0.760** | **0.266** | 20.26 | 0.716 | 0.279 |
| floating tree | **20.82** | **0.769** | **0.204** | 20.67 | 0.733 | 0.211 |
| aspen | **17.16** | **0.344** | **0.611** | 16.48 | 0.292 | 0.657 |
| stump | **22.62** | **0.820** | **0.194** | 21.96 | 0.753 | 0.213 |
| sculpture | **22.28** | **0.692** | **0.296** | 21.74 | 0.652 | 0.315 |
| Giannini Hall | **19.29** | **0.590** | **0.406** | 17.66 | 0.491 | 0.444 |
| mean | **21.54** | **0.693** | **0.311** | 20.32 | 0.619 | 0.346 |

use the GT pose, although there is still a minor gap between our method and Voxurf with the GT pose, *i.e.*, 0.85mm vs. 0.72mm.

**Results on MobileBrick.** The pose results are summarised in Tab. 3. We initialise all baseline methods, including ours, with the ARKit pose. Note that the provided GT pose in this dataset is imperfect, which is obtained by pose refinement from human labelling. The results show that our method is closer to the GT pose than others. More importantly, our method can consistently improve pose accuracy in all scenes, while other methods cannot.

The reconstruction results are presented in Tab. 4. It shows that the refined pose by all methods can lead to an overall improvement in reconstruction performance. Compared with BARF Lin et al. (2021) and SPARF Truong et al. (2023), our method shows more significant improvement. As a result, our approach surpasses the provided GT pose in terms of reconstruction F1 score, *i.e.*, 75.67 vs 75.13, achieving state-of-the-art performance on the MobileBrick dataset.

## 5.2 GENERALISATION

The proposed MLP-based pose optimisation method can be readily plugged into different NeRF-like systems to improve performance. To demonstrate the universal use of our method, we integrate it into the Nerfstudio library Tancik et al. (2023), which is a modular framework for neural radiance field development. We compare our method with the recommended baseline method (Nerfacto), which integrates multiple methods into one, including pose optimisation. For a fair comparison, we replace the original pose parameter optimisation module with our proposed PoRF MLP, and we do not use the proposed loss function. Tab. 5 provides the evaluation results on the Nerfstudio dataset. The dataset comprises ten in-the-wild, 360-degree captures obtained using either a mobile phone or a mirror-less camera with a fisheye lens. The data was processed using COLMAP or the Polycam app to obtain camera poses and intrinsic parameters. It shows that our method can consistently boost novel view rendering performance in diverse scenes.

## 5.3 ABLATION STUDIES

We conduct extensive ablation studies on the DTU dataset Jensen et al. (2014). The pose errors during training are shown in Fig. 3, where we compare six different settings and we can draw some key observations from the ablation studies:

**PoRF vs pose parameter optimisation:** L2 surpasses L1 in terms of accuracy, thereby highlighting the benefits of utilising the Pose Residual Field (PoRF) in contrast to conventional pose parameter optimisation within the same baseline framework, where solely the neural rendering loss is employed. Upon the integration of the proposed epipolar geometry loss $L_{EG}$ with the baseline approach, L4 outperforms L3, substantiating the consistency of the observed outcomes.

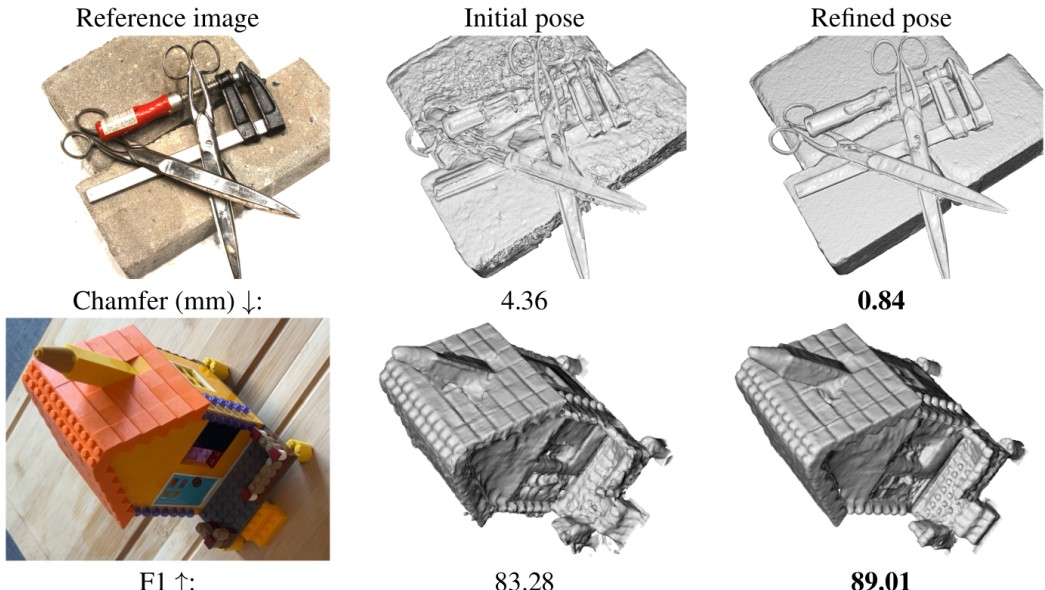

Figure 4: Reconstruction results on DTU (top) and MobileBrick (bottom) datasets. The initial pose denotes the COLMAP pose on DTU, and the ARKit pose on MobileBrick. We use the standard evaluation metrics, *i.e.*, Chamfer distance (mm) for DTU and F1 score for MobileBrick.

**Efficacy of the epipolar geometry loss:** L3 demonstrates faster convergence when compared to L1, offering substantiation of the efficacy of the proposed $L_{EG}$ loss within the context of pose parameter optimisation. Upon incorporating the proposed Pose Residual Field (PoRF), L4 surpasses L2, thus reinforcing the same conclusion.

**Ablation over correspondences:** we compare handcrafted matching methods (L4 representing SIFT Lowe (2004)) with deep learning-based methods (L5 denoting LoFTR Sun et al. (2021)). On average, SIFT yields 488 matches per image pair, while LoFTR generates a notably higher count of 5394 matches. The results indicate that both methods achieve remarkably similar levels of accuracy and convergence. It's worth noting that the SIFT matches are derived from COLMAP results, so their utilisation does not entail additional computational overhead.

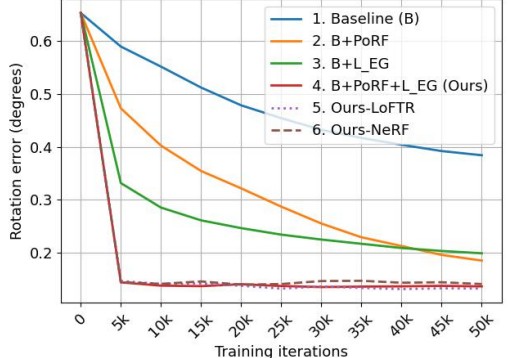

Figure 3: Pose errors during training on the DTU dataset. The results are averaged over 15 test scenes. Baseline (B) denotes the naive joint optimisation of NSR and pose parameters.

**Ablation over scene representations:** We substitute the SDF-based representation (L4) with the NeRF representation Mildenhall et al. (2020), denoted as L6. This adaptation yields comparable performance, highlighting the versatility of our method across various implicit representations.

## 6 CONCLUSION

This paper introduces the concept of the Pose Residual Field (**PoRF**) and integrates it with an epipolar geometry loss to facilitate the joint optimisation of camera pose and neural surface reconstruction. Our approach leverages the initial camera pose estimation from COLMAP or ARKit and employs these newly proposed components to achieve high accuracy and rapid convergence. By incorporating our refined pose in conjunction with Voxurf, a cutting-edge surface reconstruction technique, we achieve state-of-the-art reconstruction performance on the MobileBrick dataset. Furthermore, when evaluated on the DTU dataset, our method exhibits comparable accuracy to previous approaches that rely on ground-truth pose information.

ACKNOWLEDGEMENT

The authors gratefully acknowledge the financial support provided by Apple. This work is also supported by the UKRI grant: Turing AI Fellowship EP/W002981/1 and EPSRC/MURI grant: EP/N019474/1. We also thank the Royal Academy of Engineering and FiveAI.

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

## A    EXPERIMENTAL RESULTS

### A.1    DATASETS

The DTU Jensen et al. (2014) dataset contains a variety of object scans with 49 or 64 posed multi-view images for each scan. It covers different materials, geometry, and texture. We evaluate our method on DTU with the same 15 test scenes following previous work, including IDR Yariv et al. (2020), NeuS Wang et al. (2021a), and Voxurf Wu et al. (2023). The results are quantitatively compared by using Chamfer Distance, given the ground truth point clouds.

The MobileBrick Li et al. (2023a) contains iPhone-captured 360-degree videos of different objects that are built with LEGO bricks. Therefore, the accurate 3D mesh, obtained from the LEGO website, is provided as the ground truth for evaluation. The ARKit camera poses and the manually refined ground-truth poses by the authors are provided, however, the accuracy is not comparable to the poses obtained by camera calibration such as that in the DTU dataset. We compare our method with other approaches on 18 test scenes by following the provided standard pipeline.

### A.2    EVALUATION METRICS

To assess the accuracy of camera poses, we employ a two-step procedure. First, we perform a 7-degree-of-freedom (7-DoF) pose alignment to align the refined poses with the ground-truth poses. This alignment process ensures that the calculated poses are in congruence with the ground truth. Subsequently, we gauge the alignment's effectiveness by utilising the standard absolute trajectory error (ATE) metric, which quantifies the discrepancies in both rotation and translation between the aligned camera poses and the ground truth. Moreover, we utilise these aligned camera poses to train Voxurf Wu et al. (2023) for object reconstruction. This ensures that the reconstructed objects are accurately aligned with the ground-truth 3D model, facilitating a meaningful comparison between the reconstructed and actual objects.

### A.3    IMPLEMENTATION DETAILS

**Coordinates normalisation for unbounded 360-degree scenes.**    In the DTU Jensen et al. (2014) and MobileBrick Li et al. (2023a) datasets, images include distant backgrounds. The original NeuS Wang et al. (2021a) either uses segmentation masks to remove the backgrounds or reconstructs them using an additional NeRF model Mildenhall et al. (2020), following the NeRF++ method Zhang et al. (2020b).

In contrast, we adopt the MipNeRF-360 approach Barron et al. (2022), which employs a normalisation to handle the point coordinates efficiently. The normalisation allows all points in the scene to be modelled using a single MLP network. The normalisation function is defined as:

$$\text{contract}(\mathbf{x}) = \begin{cases} \mathbf{x} & \|\mathbf{x}\| \leq 1 \\ \left(2 - \frac{1}{\|\mathbf{x}\|}\right)\left(\frac{\mathbf{x}}{\|\mathbf{x}\|}\right) & \|\mathbf{x}\| > 1 \end{cases} \tag{10}$$

By applying this normalisation function, all points in the scene are effectively mapped to the range (0-2) in space. This normalised representation is used to construct an SDF field, which is then utilised for rendering images. Therefore, the proposed method can process images captured from 360-degree scenes without the need for segmentation masks.

**Rendering network architectures.**    Our neural surface reconstruction code is built upon NeuS Wang et al. (2021a), which means that the network architectures for both SDF representation and volume rendering are identical. To provide more detail, the SDF network comprises 8 hidden layers, with each layer housing 256 nodes, and it employs the ELU activation function for nonlinearity. Similarly, the rendering network consists of 4 layers and utilises the same nonlinear activation functions. This consistency in architecture ensures that the SDF and rendering processes are aligned within our implementation.

**PoRF MLP architectures.**    We employ a simple neural network structure for our PoRF (Pose Refinement) module, consisting of a 2-layer shallow MLP. Each layer in this MLP comprises 256

nodes, and the activation function employed is the Exponential Linear Unit (ELU) for introducing nonlinearity. The input to the PoRF MLP consists of a 7-dimensional vector. This vector is comprised of a 1-dimensional normalised frame index and a 6-dimensional set of pose parameters. The PoRF MLP's output is a 6-dimensional vector, which represents the pose residual.

**Correspondence generation.** By default, we export correspondences from COLMAP, which are generated using the SIFT descriptor Lowe (2004) and the outliers are removed by using a RANSAC-based two-view geometry verification approach. These correspondences, while reliable, tend to be sparse and may contain noise. However, our method is designed to automatically handle outliers in such data. Additionally, we conduct experiments with correspondences generated by LoFTR Sun et al. (2021). These correspondences are dense, providing a denser point cloud representation. We adhere to the original implementation's guidelines to select high-confidence correspondences and further employ a RANSAC-based technique to eliminate outliers. Consequently, the final set of correspondences obtained through LoFTR remains denser than SIFT matches.

**Training details.** Our approach follows a two-stage pipeline. In the first stage, we undertake the training of the complete framework, conducting a total of 50,000 iterations using all available images. This comprehensive training enables the refinement of pose information for each image. In the second stage, we employ Voxurf Wu et al. (2023) for the purpose of reconstruction. It's important to note that this reconstruction is executed solely using the training images. This step ensures a fair comparison with prior methods, encompassing both surface reconstruction and novel view rendering. Regarding the hyperparameters used in our joint training of neural surface reconstruction and pose refinement, we have adopted settings consistent with those detailed in NeuS Wang et al. (2021a). Specifically, we randomly sample 512 rays from a single image in each iteration, with 128 points sampled per ray.

## A.4 DETAILS FOR BASELINE METHODS

BARF Lin et al. (2021) and L2G Chen et al. (2023) are designed for LLFF and NeRF-Synthetic datasets. We find that naively running them with the original setting in DTU and MobileBrick scenes leads to divergence. Therefore, we tuned hyper-parameters for them. Specifically, for BARF, we reduce the pose learning rate from the original 1e-3 to 1e-4. For L2G, we multiply the output of their local warp network by a small factor, *i.e.*, $\alpha = 0.01$. As SPARF Truong et al. (2023) was tested in DTU scenes, we do not need to tune parameters. However, as it runs on sparse views (e.g., 3 or 6 views) as default, they build all-to-all correspondences. When running SPARF in dense views (up to 120 images), we build correspondences between neighbouring co-visible views (up to 45 degrees in the relative pose) considering the memory limit.

## A.5 DETAILS FOR EXPERIMENTAL RESULTS

**Quantitative ablation study results on DTU.** Tab. R1 shows the quantitative results for different settings illustrated as curves in Fig. 3. Firstly, the comparison spanning from L1 to L4 serves to validate the effectiveness of the proposed components, which encompass critical elements like the PoRF (Pose Refinement) module and the epipolar geometry loss. This analysis highlights the significance of these components in enhancing the overall performance of the method. Secondly, the comparison encompassing L4 to L6 is intended to showcase the versatility and adaptability of the proposed method across different correspondence sources and scene representations. This demonstration underlines the method's capacity to deliver robust results regardless of variations in input data, thereby emphasising its universal applicability.

Table R1: Quantitative ablation study results on DTU. The results are averaged over 15 test scenes. Baseline (B) denotes naive joint optimisation of pose parameters and NSR.

| Methods | COLMAP | 1. Baseline (B) | 2. B+PoRF | 3. B+$L_{EG}$ | 4. B+PoRF+$L_{EG}$ (Ours) | 5. Ours-LoFTR | 6. Ours-NeRF |
|---|---|---|---|---|---|---|---|
| Rot. (Deg.) | 0.65 | 0.38 | 0.19 | 0.20 | 0.14 | **0.13** | 0.14 |
| Tran. (mm) | 1.03 | 2.67 | 1.06 | 1.37 | 1.02 | 1.03 | **1.00** |

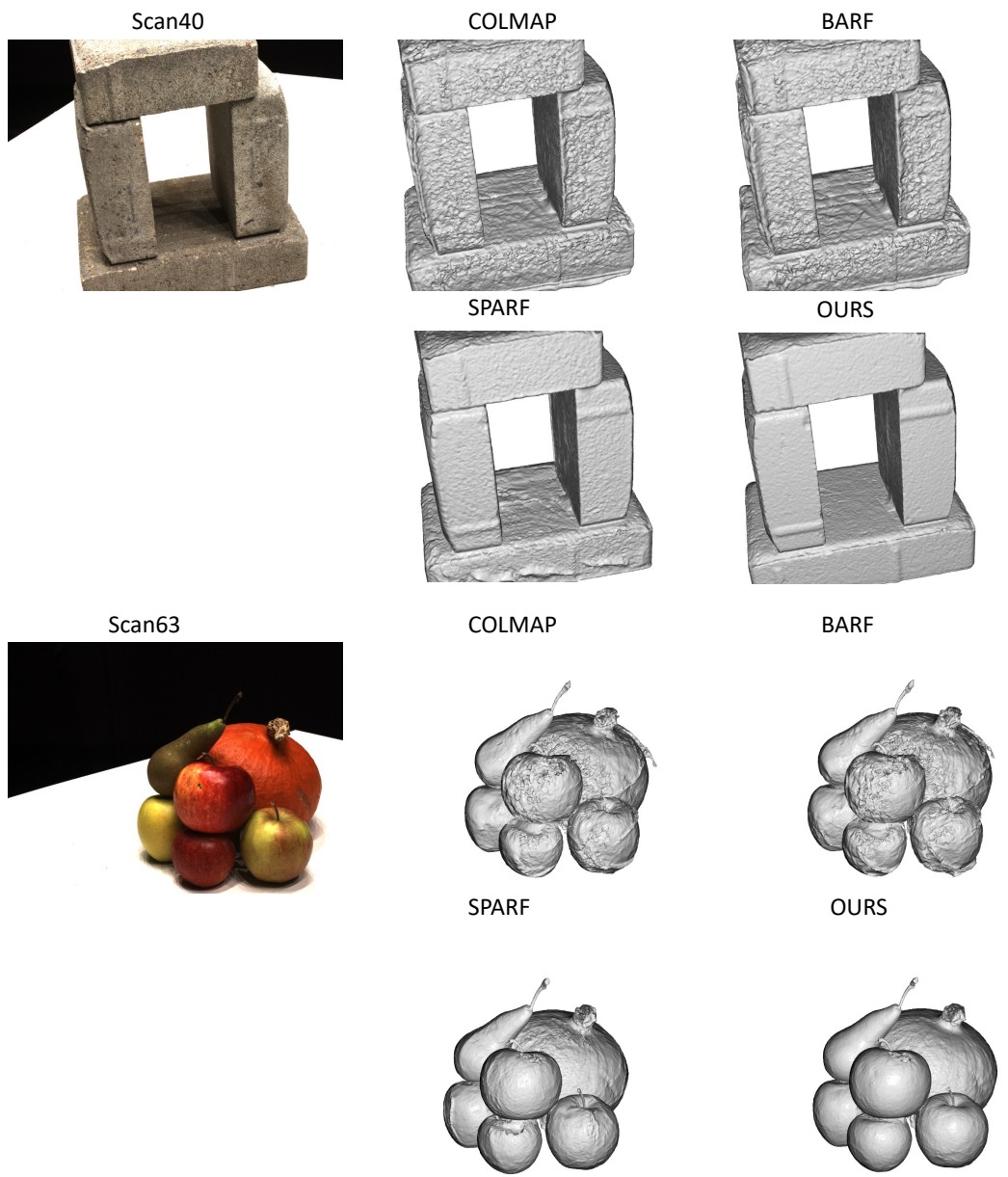

Figure S1: Qualitative reconstruction results on the DTU dataset. All meshes were generated by using Voxurf Wu et al. (2023). All refinement method takes the COLMAP Schönberger & Frahm (2016) pose as the initial pose.

**Qualitative reconstruction comparison.** We include additional qualitative reconstruction results for both the DTU dataset in Fig. S1 and the MobileBrick dataset in Fig. S2. These visualisations serve to illustrate key points. It is evident that the reconstruction quality using COLMAP Schönberger & Frahm (2016) pose information is marred by noise, making it challenging to enhance even when employing BARF Lin et al. (2021). SPARF Truong et al. (2023) demonstrates improvements in visual quality, but it lags behind our method in terms of the precision and fidelity of the reconstructed object surfaces. These supplementary qualitative results underscore the superior performance and capabilities of our method in comparison to alternative approaches, particularly when it comes to achieving accurate and detailed object surface reconstructions.

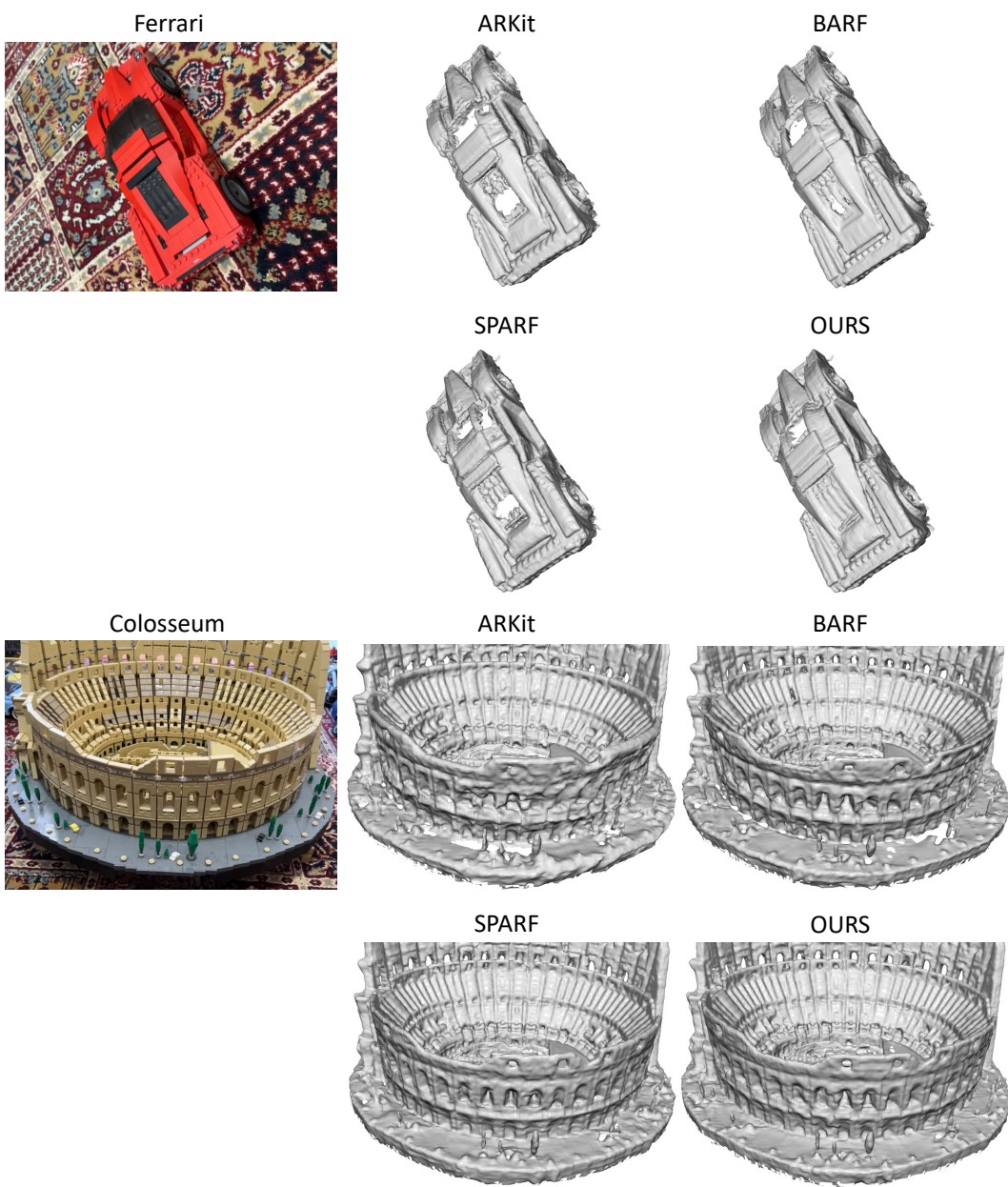

Figure S2: Qualitative reconstruction results on the MobileBrick dataset. All meshes were generated by using Voxurf Wu et al. (2023). All refinement method takes the ARKit pose as the initial pose.

## B ADDITIONAL EXPERIMENTAL RESULTS

### B.1 NOVEL VIEW SYNTHESIS

We have also provided additional novel view synthesis results on both the DTU dataset (Table R2) and the MobileBrick dataset (Table R3). In these evaluations, we utilise Voxurf Wu et al. (2023) for image rendering. It's essential to note that Voxurf was not originally designed for image rendering purposes, and it may face challenges, particularly when dealing with images that contain distant backgrounds, as observed in the MobileBrick dataset. These additional results offer a comprehensive view of our method's performance across different datasets, emphasising its strengths and limitations

in novel view synthesis tasks. The qualitative comparison results are illustrated in Fig. S6 and Fig. S7, respectively.

Table R2: Quantitative novel view synthesis results on the DTU dataset. The Voxurf Wu et al. (2023) is used for reconstruction and image rendering.

| Poses | GT | COLMAP | BARF Lin et al. (2021) | SPARF Truong et al. (2023) | OURS |
|---|---|---|---|---|---|
| PSNR ↑ | 31.84 | 27.63 | 27.50 | 29.59 | **31.67** |
| SSIM ↑ | 0.931 | 0.861 | 0.856 | 0.892 | **0.929** |
| LPIPS ↓ | 0.142 | 0.208 | 0.211 | 0.167 | **0.143** |

Table R3: Quantitative novel view synthesis results on the MobileBrick dataset. The Voxurf Wu et al. (2023) is used for image rendering. The images in this dataset encompass 360-degree distant backgrounds, and it's worth noting that Voxurf was not specifically designed to render images and to handle challenges in such scenarios.

| Poses | GT | ARKit | BARF Lin et al. (2021) | SPARF Truong et al. (2023) | OURS |
|---|---|---|---|---|---|
| PSNR ↑ | 23.65 | 17.94 | **19.45** | 18.87 | 18.97 |
| SSIM ↑ | 0.719 | 0.561 | **0.634** | 0.607 | 0.619 |
| LPIPS ↓ | 0.447 | 0.541 | **0.501** | 0.506 | **0.501** |

## B.2 ADDITIONAL ABLATION STUDIES

**Ablation over the inputs of PoRF.** In our PoRF module, the inputs comprise both the frame index and the initial camera pose. In this experiment, we conducted an ablation study to dissect the impact of each component on the results. Figure S3 presents the outcomes of this study, demonstrating that the removal of the initial camera pose results in a significant drop in performance, while the removal of the frame index causes a minor decrease. It's crucial to note that although the frame index appears to have a relatively low impact on accuracy, it remains an essential input. This is because the frame index is unique to each frame and serves a critical role in distinguishing frames that are closely situated in space.

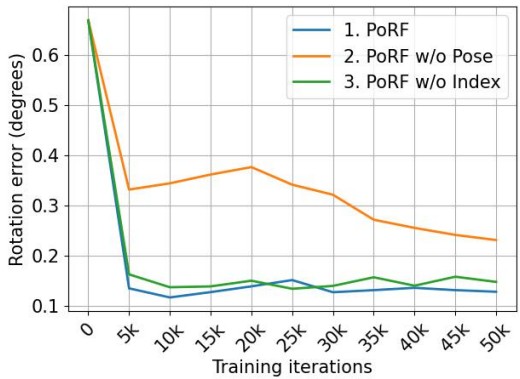

Figure S3: Ablation study results of the PoRF inputs on *Scan37*.

## B.3 ADDITIONAL ANALYSIS

**Impact of the pose noise on reconstruction.** To understand how pose noises affect reconstruction accuracy, we make an empirical analysis by introducing different levels of pose noise on DTU Jensen et al. (2014). The results are shown in Fig. S5, where we use Voxurf Wu et al. (2023) to do reconstruction with different poses. To facilitate a comparison between rotation and translation, we use the COLMAP pose error as a reference, *i.e.*, using its pose error as a unit. It reveals that reconstruction accuracy is highly sensitive to rotation errors, whereas small perturbations in translation have minimal impact on reconstruction accuracy. Therefore, in the optimisation process, the reconstruction loss cannot provide strong supervision for optimising translation. This explains why our method shows significant improvement in rotation but minor improvement in translation.

**Robustness to the initial pose noise.** To assess the robustness of our method, we introduce varying degrees of noise to the ground truth (GT) pose during initialisation and subsequently apply our pose refinement process. The outcomes of this analysis are illustrated in Fig. S4, displaying quantitative results for the DTU dataset Jensen et al. (2014). In line with the preceding analysis, we utilise the COLMAP pose error as a reference point for comparison. The findings underscore that our method

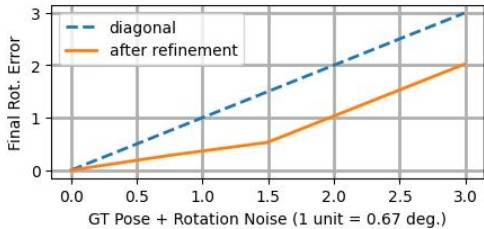 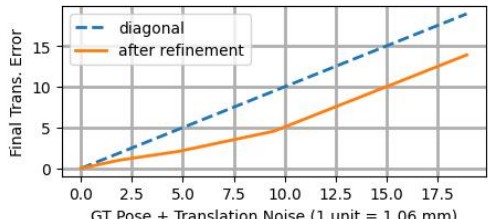

Figure S4: Robustness of our method to the initial pose noise on DTU (*Scan37*). The dashed lines and solid denote the pose error before and after refinement, respectively.

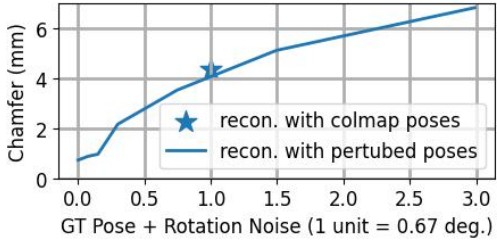 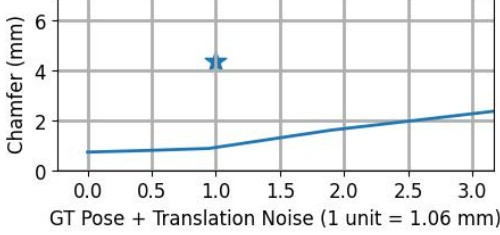

Figure S5: Reconstruction errors with pose errors on DTU (*Scan37*). We add noise to the GT pose and adopt the COLMAP pose error as one unit for comparison.

consistently reduces pose error across different levels of pose noise, revealing its capacity for robust performance. Additionally, the method exhibits greater resilience to translation errors in comparison to rotation errors.

## C LIMITATIONS

First, our method builds upon the foundation of an existing technique, NeuS Wang et al. (2021a), for the joint optimisation of neural surface reconstruction and camera poses. Consequently, the training process is relatively slow, taking approximately 2.5 hours for 50,000 iterations. In future work, we plan to explore techniques like instant NGP Müller et al. (2022) methods to accelerate this training phase further.

Second, it's essential to note that our method, unlike SPARF Truong et al. (2023), is primarily designed for high-accuracy pose refinement. As a result, it requires nearly dense views and a robust camera pose initialisation for optimal performance. In practical real-world applications, achieving such dense views and quality camera pose initialisation can often be accomplished using existing methods like ARKit and COLMAP.

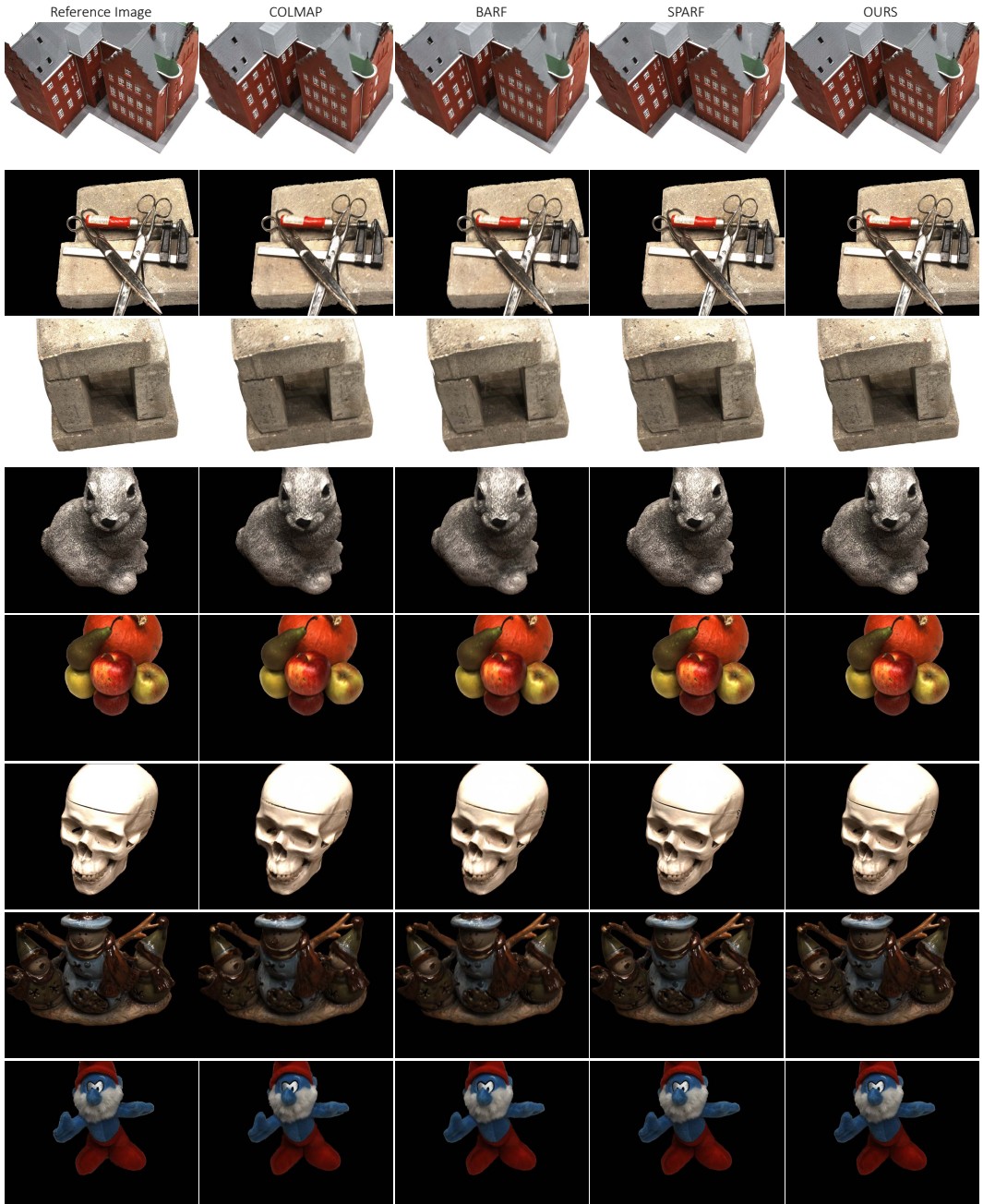

Figure S6: Qualitative novel view synthesis results on the DTU dataset. Images were rendered by using Voxurf Wu et al. (2023).

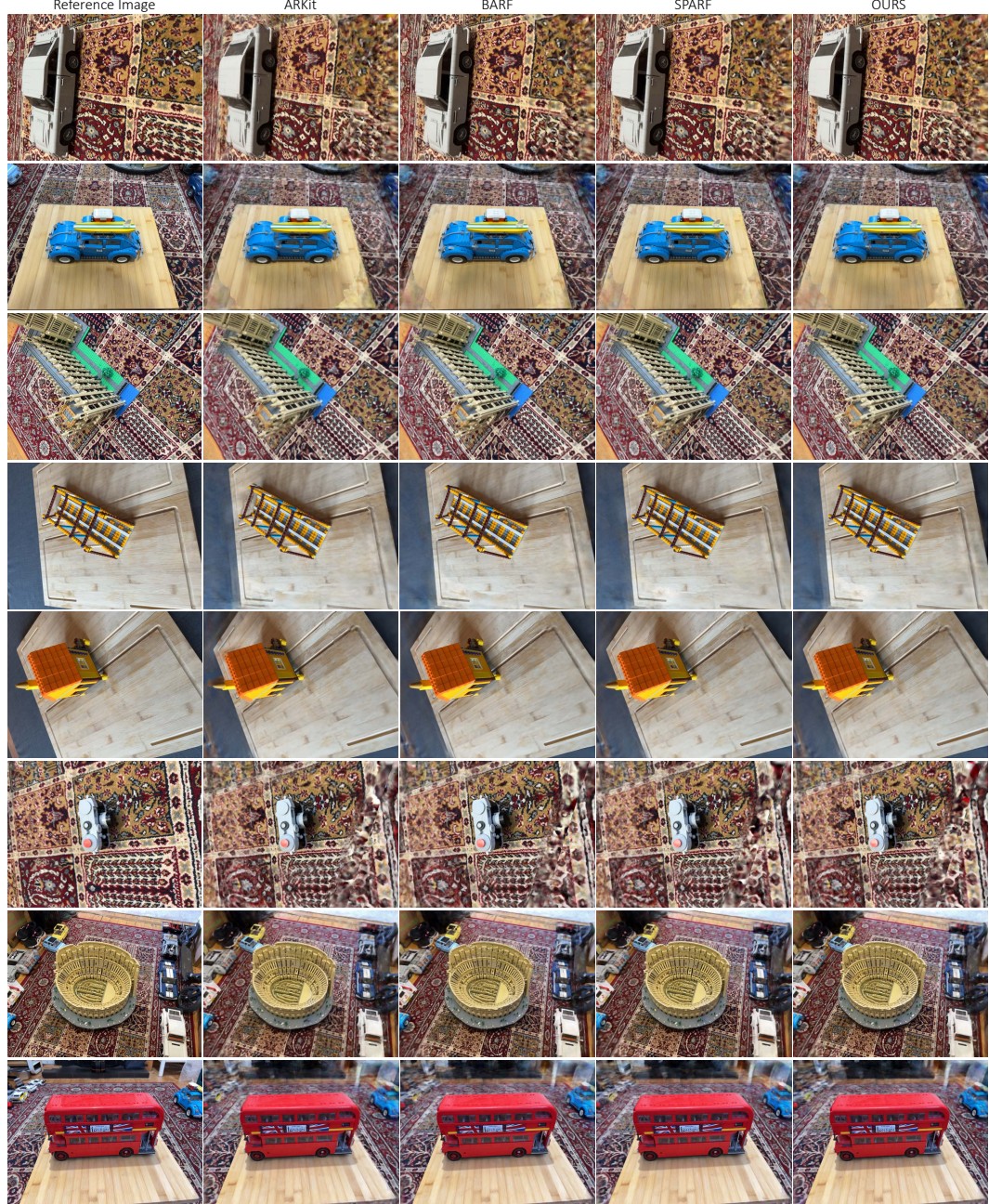

Figure S7: Qualitative novel view synthesis results on the MobileBrick dataset. Images were rendered by using Voxurf Wu et al. (2023).

