# OpenReview forum: "PORF: POSE RESIDUAL FIELD FOR ACCURATE NEURAL SURFACE RECONSTRUCTION"
_ICLR.cc/2024/Conference — ICLR 2024 poster_

### Official Review · Reviewer_dcYV · 2023-10-25

**Soundness:** 3 good
**Presentation:** 3 good
**Contribution:** 3 good
**Rating:** 6
**Confidence:** 5

**Summary:**

This paper proposes an approach called PoRF for robust neural surface reconstruction with noisy camera pose initialization. The method leverages an MLP for regressing pose updates, which is more robust and accurate than conventional pose parameter optimization. The proposed method also includes an epipolar geometry loss to enhance supervision, which leverages correspondences exported from COLMAP results without limited computational overhead. The paper presents experiments on the DTU and MobileBrick datasets, demonstrating the efficacy of the proposed approach in refining camera poses and improving the accuracy of neural surface reconstruction in real-world scenarios.

**Strengths:**

1. The proposed method is claimed to be more robust than conventional methods due to parameter sharing that leverages global information over the entire sequence.
2. The paper presents extensive experiments on the DTU and MobileBrick datasets, demonstrating the efficacy of the proposed approach in refining camera poses and improving the accuracy of neural surface reconstruction in different scenarios.
3. This paper is well-written and easy to understand, with clear explanations of the proposed method and experimental results.

**Weaknesses:**

1. It is a little unclear to me why using a single MLP for pose regression can improve pose optimization performance. Did you try to use multiple MLPs (one MLP for one frame)? Although it is claimed that one single MLP can leverage the global context information, it is not very convincing to me.
2. Why should your method even outperform NeuS in Table 2 with GT poses? Does it mean your optimized pose is better than GT's or GT poses are also erroneous?
3. The proposed method was only evaluated on two ideal object-centric datasets, which cannot demonstrate the generalization ability to different types of real-world data. I suggest the authors consider testing their performance on datasets like KITTI and mipnerf 360's dataset.
4. In Eq. (7), the square of the inlier rate is used to weigh the loss terms. Why did you just simply use squares rather than other forms, e.g., cubic? Is this just an intuitive design or a careful selection after experiments?

**Questions:**

Please refer to the weakness part.

---

> ### Author Response · Authors · 2023-11-13
> **Reviewer dcYV**
>
> Thank you for your positive comments, including the recognition that the proposed method is more robust than conventional pose optimization, the results are extensive, and the paper is well-written. We primarily address your concerns below.
>
>  [Single MLP VS multiple MLPs.]  A single MLP can learn the distribution of the camera poses over the entire sequence, which helps correct pose errors. Using multiple MLPs (one MLP per image) simply increases trainable parameters compared to traditional pose optimization. Additionally, it is impractical in a large scene (e.g., thousands of images or more) due to memory and computing limitations. Following your suggestions, we conducted an ablation study on the DTU (scan37) dataset. The results show that direct pose optimization leads to a 0.26-degree rotation error, a single MLP (PoRF) leads to a 0.13-degree error, and Multiple MLPs lead to a 0.24-degree error. This validates our claim that parameter sharing contributes to improved pose optimization.
>
> [Outperform NeuS with GT poses]. The Ground Truth (GT) poses on the DTU dataset (Table 2) are accurate, obtained through multi-camera calibration. Conversely, the GT poses on the MobileBrick dataset (Table 4) are erroneous. Our method exhibits a slightly worse performance than NeuS with GT poses (0.85 vs. 0.77 in Chamfer Distance) in Table 2, and it outperforms NeuS with GT poses (75.67 vs. 73.74 in F1 measure) in Table 4.
>
> [Test on more real-world datasets] Thank you for your suggestions. We conducted additional experiments on a diverse real-world dataset after the submission. Specifically, we integrated our PoRF into the **Nerfstudio** library [1] and performed ablation studies with the built-in state-of-the-art baseline method (Nerfacto) on the Nerfstudio dataset. Note that the "Nerfstudio Dataset" comprises 10 in-the-wild captures obtained using either a mobile phone or a mirror-less camera with a fisheye lens. The data was processed using COLMAP or the Polycam app to obtain camera poses and intrinsic parameters. This dataset offers researchers more than 360 real-world captures not limited to forward-facing scenes. It is similar to MipNeRF-360 but does not focus on a central object and includes captures with varying degrees of quality. The results (PSNR, the higher the better) are summarized in the following table.
>
> |               | Egypt     | person    | kitchen   | plane     | dozer     | floating-tree | aspen     | stump     | sculpture | Giannini-Hall |
> |---------------|-----------|-----------|-----------|-----------|-----------|---------------|-----------|-----------|-----------|---------------|
> | Nerfacto      | 20.06     | 23.37     | 20.11     | 20.96     | 19.98     | 19.24         | 16.31     | 21.89     | 20.14     | 18.68         |
> | Nerfacto-PoRF | **22.64** | **25.32** | **20.28** | **21.87** | **20.45** | **19.91**     | **16.72** | **23.13** | **21.44** | **19.18**     |
>
>
> Note that Nerfacto contains the built-in pose parameter optimization method, similar to BARF and NeRFmm. We simply replaced this module with our proposed PoRF (we did not use the proposed epipolar geometry loss here for a fair comparison with the baseline), and it consistently improved performance in all 10 scenes. This demonstrates that our method is generalizable to various real-world scenarios.
>
> [Square weights in Eq 7]  We have tested multiple weight choices, including linear, square, cubic, and exponential ones. The results show that they have very minor differences, and we finally chose square weights for their simplicity and intuitiveness.
>
> [1] Nerfstudio: A Modular Framework for Neural Radiance Field Development, Tancik et al., ACM SIGGRAPH 2023 Conference Proceedings, 2023

---

> > ### Comment · Reviewer_dcYV · 2023-11-22
> >
> > Thanks for addressing my raised issues.

---

### Official Review · Reviewer_jgNN · 2023-10-31

**Soundness:** 3 good
**Presentation:** 3 good
**Contribution:** 1 poor
**Rating:** 5
**Confidence:** 4

**Summary:**

This paper targets the optimization of camera pose for neural surface reconstruction. For handling the camera pose noise and intensive-view scenarios, this paper introduces PoRF, which learns the pose residuals (i.e., offset) by the frame index and initial camera pose. Moreover, the epipolar geometry loss is used to enhance supervision for camera pose estimation.  Along with the volume rendering and reconstruction loss in NeuS as the baseline, the proposed method shows high accuracy and robustness in intensive-view scenarios and a noisy initial camera pose. The proposed method is validated on the DTU and MobileBrick datasets, respectively. It uses the COLMAP and ARKit poses as the initial camera poses and achieves better camera poses and reconstruction results compared to other SOTA.

**Strengths:**

- The proposed method is simple yet effective for joint optimization of camera pose and neural surface reconstruction.
- The results are impressive. Tabs. 1-4 show the proposed method achieves high-quality camera pose estimation and neural surface reconstruction. Moreover, Figs. 3-5 show the proposed method is robust to noise.
- The paper is well organized and easy to follow.

**Weaknesses:**

[Fig.2] Based on Eq.6, $\alpha$ should multiply the residuals. However, Fig.2 shows $\alpha$ multiply the input.

[Ablation of PoRF]
- [Shared MLP] The paper claims that the shared MLP captures the underlying global information and therefore boosts performance. However, this is not discussed in Fig.S5 or ablation study.
- [$\alpha$] It would be better to discuss the use of $\alpha$ like w/ and w/o the fixing factor $\alpha$ and its value. This is not discussed in Fig.S5.

[Design of PoRF] The PoRF formulates camera pose estimation as a pose refinement procedure, which takes initial poses as input and predicts the offsets. The final prediction is the initial poses plus the offsets. This paper claims the PoRF was inspired by Nerf and can be treated as one of its main contributions. However, similar modules like recurrent pose refinement or iterative offset regression are a standard module in pose estimation, including 6D object[1,4] or human pose estimation[2,3]. Compared to existing modules, PoRF only introduces an additional index as input. This makes the novelty incremental. It would be better to highlight the difference or advantage of PoRF compared to those existing modules.


[Epipolar] Similar Epipolar geometry contraints are also used in multi-view tasks like [5,6]. They all extract keypoints and then use Sampson distance. The framework is like a combination, which limits its novelty.


References
- [1] CRT-6D: Fast 6D Object Pose Estimation with Cascaded Refinement Transformers. WACV2023.
- [2] End-to-end Hand Mesh Recovery from a Monocular RGB Image. ICCV2019.
- [3] Human Pose Estimation with Iterative Error Feedback. CVPR2016.
- [4] RNNPose: Recurrent 6-DoF Object Pose Refinement with Robust Correspondence Field Estimation and Pose Optimization. CVPR2022.
- [5] Deep Keypoint-Based Camera Pose Estimation with Geometric Constraints. IROS2020.
- [6] PoseDiffusion: Solving Pose Estimation via Diffusion-aided Bundle Adjustment

**Questions:**

See Weaknesses. My main concern is the novelty.

---

> ### Author Response · Authors · 2023-11-13
> **Reply to Reviewer jgNN**
>
> Thank you for your positive comments, including the recognition that the proposed method is simple yet effective, the results are impressive, and the paper is well-written. We primarily address your concerns, particularly regarding the novelty of this paper below.
>
> [Novelty of PoRF] We will discuss the mentioned papers in the final version, however, we argue that [1-4] addresses the generic pose-prediction problem across scenes, while our method focuses on per-scene pose optimization. The differences are:
>
> -   [1-4] follows a "training-inference" paradigm, which is a standard machine learning approach trainable on a large dataset, predicting results (camera poses or pose offsets) during inference in new scenes. The pose network conditions on generalizable feature embeddings extracted from image appearance or geometry information like correspondences or optical flow.
>
> -   Our PoRF is a per-scene optimization method, like NeRF. Our method is randomly initialised in each scene and optimizes camera pose parameters for all frames. Thus, it can take frame indices as the only input—refer to "PoRF-w/o pose” in Fig. S5. Noteworthy related works include BARF, SPARF, and Nope-NeRF, which directly optimize camera pose parameters (or pose offsets) for each frame.
>
> Furthermore, recurrent pose refinement and iterative offset optimization have historical roots in classical multi-view geometry methods such as Visual SLAM and Structure-from-Motion systems, which are also predating [1-4]. Here, our paper asserts novelty as the first to leverage an MLP-based solution for camera pose refinement in the context of NeRF and NeuS.
>
> [Novelty of Epipolar Loss] The use of epipolar geometry constraints has a historical foundation in 3D computer vision and multi-view geometry. We contend that our application of epipolar geometry to the novel domain of NeRF problems constitutes a unique contribution. Please refer to our response to Reviewer 7s1S for more discussion on the novelty of our epipolar loss.
>
> [Fig. 2] Apologies for any confusion; we will revise Fig 2 in the final version for clarity.
>
> [Discussion of PoRF Parameter Sharing] In response to Reviewer dcYV's suggestion, we added an ablation study where we employed Multiple MLPs (one MLP for each image), eliminating parameter sharing. The results indicate that our single MLP solution exhibits significantly better performance. Please refer to our reply to Reviewer dcYV for more details.
>
> [Alpha] The alpha parameter is used to scale network outputs. A small network output can keep the refined pose close to the initial pose, crucial for optimizing networks at the beginning. With the training, the network can always learn to predict results at the correct scale, so fixing or varying alpha yields comparable results. In this paper, we fix it for simplicity. Moreover, we find that it works well when alpha is smaller than 1e-2, which shows similar results within the range of 1e-6 to 1e-2.

---

> > ### Comment · Reviewer_jgNN · 2023-11-20
> >
> > Thanks for the feedback, and I appreciate the performance and the simple design. Most of my concerns are addressed, except for novelty.
> >
> > I agree that this paper is a follow-up of Nope-NeRF for camera pose estimation in the context of NeRF. Note that existing works like Nope-NeRF have well-established joint optimization of camera poses and NeRF. Especially, Nope-NeRF introduces the concepts of **relative poses** and **matching losses** as two contributions. This paper follows Nope-NeRF and introduces **Pose Residual Field (PoRF)** and **epipolar geometry loss** correspondingly. With the similar overall framework and the point-to-point improvements, I think this further increases the requires of the novelty. This is the reason that I asked for a high-level novelty standard for the two modules.
> >
> > - I agree that **recurrent pose refinement or iterative offset optimization** has historical roots. They are commonly designed for optimization methods very early, and then for data-driven methods or hybrid methods like deep learning recently. In these aspects, Nerf can also be treated as optimization-based. So it is not convincing that the PoRF is novel because [1-4] are data-driven-based.
> >
> > - Especially, PoRF is similar to the strategy "render-and-compare and pose offset update" like [4,7] which commonly designed for 6D pose estimation. For those works, as the input is initial poses and the output is residual, MLP is their straightforward design.
> >
> > - Random initialization is the advantage of offset optimization for optimization methods. But initialization from data-driven-based methods makes the performance better, and this is the motivation for hybrid methods.
> >
> > - I think the contribution of PoRF is over-claimed. Camera pose estimation is also a kind of pose estimation. The overlook of existing common pose estimation works makes the claim not convincing. Instead, I do not think PoRF is an implicit pose representation. For example, in the introduction, the paper highlights their design as implicit pose representation and their shared MLP network. In the method, they claim PoRF is motivated by coordinates-MLP and Nerf.
> >
> > - I agree that this paper first introduces the concept of residual for Nerf. But from the aspect of novelty, I am concerned with the contribution how to extend an existing concept to Nerf. At this time, it seems the only difference is an additional index. And, in B.2, it seems that the index is not important and just for distinguishing frames. "It’s crucial to note that although the frame index appears to have a relatively low impact on accuracy, it remains an essential input."
> >
> > - For the novelty of epipolar geometry loss, I will discuss with Reviewer 7s1S.
> >
> > [7] DeepIM: Deep Iterative Matching for 6D Pose Estimation

---

> > > ### Author Response · Authors · 2023-11-20
> > > **Reply to Reviewer jgNN**
> > >
> > > Thanks for your feedback. We appreciate the reviewer for mentioning related papers, and we will discuss them [1-7] in the final version, however, we argue that our method exhibits clear differences from all previous methods in both concepts and implementations. We provide a reply to clarify the novelty below.
> > >
> > > - [Novelty in high-level concepts] Our PoRF is the first network-based pose optimisation method in NeRF and NeuS. Our method is the only one to improve on traditional approaches (COLMAP) and outperforms even “GT poses” in challenging scenarios. Our method is simple, but we see that as a feature because it is more likely to be adopted by others. Our method is novel due to the following reasons:
> > >
> > >   - Existing pose-nerf joint optimisation methods [BARF, Nope-NeRF, etc] directly optimize per-frame pose parameters, which is similar to traditional methods such as COLMAP, while our method uses MLP for pose refinement.
> > >
> > >   - Existing network-based pose estimation methods [1-4] [7] do not relate to NeRF and 3D reconstruction. They address the generic pose estimation problem (for one image or a pair of images), while our method solves a Bundle Adjustment problem, estimating camera poses for all frames of a scene.
> > >
> > >   - More specifically, [1-4] [7] are data-driven methods and hence require additional features as input. For example, [4] uses correspondences and [7] uses image features. In contrast, our PoRF is an optimisation-based method and can take the frame index only as input.
> > >
> > > - [Motivated by Coordinates-MLP and NeRF]
> > >
> > >   -  PoRF uses the Corrdinates-MLP, which is similar to NeRF. Specifically, the input of PoRF can be viewed as time coordinates (frame index) and space coordinates (camera location and orientation). We hope this should be enough to back our claim of  “motivated by Coordinates-MLP and NeRF”.
> > >
> > >   -  NeRF is commonly viewed as an implicit scene representation because it encodes scene geometry and colours in an MLP. Similarly, our method encodes pose (residual) in an MLP, so we view our method as an implicit pose representation.
> > >
> > > - [Frame index] The frame index is a necessary input to our method, which identifies each frame. Indeed, the initial pose is an optional input to PoRF, although it can improve performance significantly. In the scenario when the initial pose is unavailable, our method can work well with the frame index only — "PoRF-w/o pose” in Fig. S5.

---

> > > > ### Comment · Reviewer_jgNN · 2023-11-22
> > > >
> > > > Thanks for the discussion.
> > > >
> > > > - From the optimizer aspects, the proposed PoRF belongs to learning to optimize [9,10,11], which has been widely used for pose estimation. **All the characteristics of PoRF like MLP, residual, init, and network-based, are due to the learning to optimize strategy for pose estimation.** Especially, existing works prefer hybrid. Because the init is important to the optimization and they also need generalizations for the data. This is a difference, not an advantage. The proposed PoRF also uses COLMAP as init. Indeed, starting with a random init for learning to optimize is OK but will lead to a performance drop. From my understanding, it is not straightforward to claim PoRF is a novel implicit representation instead of standard learning to optimize. Therefore, I suggest discussing PoRF from the optimizer aspects for pose estimation in the paper instead of Coordinates-MLP and NeRF.
> > > >
> > > > - For application, similar architecture and learning strategy can be found in other related pose estimation works [1-4,7-8], which can be applicable to camera pose estimation of Nerf if adding frame index. Therefore, I suggest highlighting the contribution of applying to Nerf in the paper.
> > > >
> > > > - I appreciate the performance. But at this time, I still think the overall novelty is limited.
> > > >
> > > >
> > > > [8] Learned Vertex Descent: A New Direction for 3D Human Model Fitting
> > > >
> > > > [9] Discriminative Optimization: Theory and Applications to Computer Vision
> > > >
> > > > [10] Understanding and correcting pathologies in the training of learned optimizers
> > > >
> > > > [11] Learning to Optimize: A Primer and A Benchmark

---

> > > > > ### Author Response · Authors · 2023-11-22
> > > > > **Reply to Reviewer jgNN**
> > > > >
> > > > > We thank the reviewer for taking the time to message us about the paper. The papers cited by the reviewer [1-4, 7, 8] learn to regress a pose from an image using a neural network -- this is a common problem in computer vision and has been dealt with by the papers cited by the reviewer and by many others (e.g. the entire body of relocalisation work).
> > > > >
> > > > > Our paper, in contrast, effectively learns a *scene-specific* neural network that *finds, and memorises, pose corrections* to fit the final nerf / implicit representations. We are happy to note the learning to optimise paradigm in our paper. It is however quite different, in that ours is a model refinement / fitting / memorisation problem, whereas the papers cited by the reviewer aim to be generic pose regression solutions. This also places our work firmly in the line of pose optimisation for scene-specific NeRF / implicit representation papers such as BARF, NeRF--, GARF, Nope-NeRF.
> > > > >
> > > > > Furthermore, akin to NeRF memorising geometry within a neural network, our method memorises pose offsets within a neural network -- this motivates our desire to call ours an implicit pose representation. Our neural network architecture is indeed not especially novel (e.g. an MLP with a skip connection, so again, not dissimilar from a nerf), but the various other contributions (different loss, index, etc) are novel and lead to state-of-the-art results.
> > > > >
> > > > > To summarise, we will happily cite the papers noted by the reviewer and discuss the "learning to optimise" paradigm, but we will note that we are finding and memorising pose offset rather than regressing like [1-4, 7, 8].

---

### Official Review · Reviewer_7mvY · 2023-10-31

**Soundness:** 3 good
**Presentation:** 4 excellent
**Contribution:** 4 excellent
**Rating:** 8
**Confidence:** 4

**Summary:**

The manuscript introduces the use of an additional small MLP to refine the poses given to neural reconstruction methods jointly with the standard neural reconstruction objective. An additional epipolar loss derived from sparse feature matching (like SIFT matches from COLMAP) is used to effectively supervise this residual pose network.
This small addition if a learned shared pose residual update has big impacts on the quality of the reconstruction as demonstrated in the experiments. Notably it outperforms considerably the direct optimization of the poses with the standard reconstruction loss formulation.

**Strengths:**

The addition of the pose residual learning with a small MLP is a straight forward addition to any neural representation learning method.
The fact that sparse correspondences needed for the epipolar loss are available from COLMAP without additional cost is an added benefit that makes it easy to incorporate into existing methods.

The quantitative experiments are high quality and compare the proposed approach against a diverse set of existing methods. They clearly show the improvements in reconstruction quality and pose estimation.

The additional ablations wrt to different algorithm settings are illustrative and help clarify that both PORF and epipolar line loss are needed to achieve high performance (Fig 5). As well as that the method can be used together with NERF (instead of just NeuS) and that SIFT matches are sufficient for the epipolar loss (vs more expensive deep learned LOFTR).

The qualitative results in Fig 1 and 6 very clearly show the ability of the poses residuals to correct for very noisy surfaces.

The manuscript is clear and well written and easy to follow except some details (see weaknesses).

**Weaknesses:**

The use of L1-L6 is unclear and not well introduced. To my knowledge this notation is not common to denote different experiments or ablations. I highly recommend introducing it upfront or using short acronyms to denote different configs. This would improve readability of the manuscript.

The effects of training the additional MLP and computing the additional epipolar loss is likely not significant when compared to training without them but it would still be good to quantify any difference in terms of overall time to convergence. If this additional time is negligible it could further support the use of PORF as a no-brainer.

Its a minor thing but I am curious as to how much the time/frame id input is used by PORF. Ablating with and without time input would show whether PORF mostly compensates for global shifts or whether it can learn to fix individual poses.

**Questions:**

The axis angle formulation for pose is a common way to parameterize updates to poses. It would be good to also clarify how the algorithm goes back from these to the full 4x4 (or 3x4) pose matrix used for transformations in the rest of the method.

In figure 4 there is a clear point at which the errors start growing linearly where before they were sub-linear. I am curious why the authors think this happens.

Is any position encoding used for the time/frame id input to PORF?

---

> ### Author Response · Authors · 2023-11-13
> **Reply to Reviewer 7mvY**
>
> Thank you very much for your positive feedback on our paper, acknowledging the novelty of our approach, high performance, and excellent paper writing. Below, we address your concerns and respond to your questions.
>
>
> **Reply to Weakness:**
>
> -   The use of L1-L6 is unclear. Thank you for your valuable suggestions. We will revise it and provide a clear description in the camera-ready version.
>
> -   Training time with and without PoRF. We appreciate your mention of this crucial point. Our method takes 2h35min to train the model with direct pose optimization in 50k iterations on a single A40 GPU, and it takes 2h40min to train the model with PoRF optimization. This demonstrates that PoRF introduces minor additional overhead.
>
> -   Ablation with and without time/frame id input. The results are in the supplementary material—refer to Section B2 and Fig. S5. We believe the frame index is a necessary input to the MLP as it allows each frame to be uniquely identified. For instance, when two frames are captured at very similar positions (the initial pose is very similar), the frame index can be used to distinguish them.
>
> **Reply to Questions:**
>
> -   Conversion between axis angle and rotation matrix. We utilize the function in the Kornia library, specifically, "kornia.geometry.conversions.rotation_matrix_to_axis_angle(rotation_matrix)" and "kornia.geometry.conversions.axis_angle_to_rotation_matrix(axis_angle)." Further implementation details will be provided in the camera-ready version.
>
> -   Two linear curves in Fig 4. Our method is dependent on the initial poses. When the initial pose error is below a certain threshold, our method can be very accurate. However, if it exceeds a certain threshold, our method can only marginally improve accuracy.
>
> -   Position encoding. We experimented with position encoding, but its use did not distinctly enhance overall performance. Therefore, we opted not to include it in the current version for simplicity.

---

> > ### Comment · Reviewer_7mvY · 2023-11-22
> >
> > Thank you for addressing my questions.
> >
> > > Training time with and without PoRF.
> >
> > please add to camera ready
> >
> > > Two linear curves in Fig 4.
> >
> > of course! I just really wonder why those thresholds exist and if they are a function of the dataset or the method?
> > Is there a point of catastrophic failure? I.e. what happens at way larger noise?

---

> > > ### Author Response · Authors · 2023-11-23
> > > **Reply to Reviewer 7mvY**
> > >
> > > Thanks for your reply. Of course, we will consider all questions and discussions in this rebuttal when preparing the camera-ready version.
> > >
> > > [Fig. 4]
> > > - We find that our method can generally reduce pose error on different levels of noise. We add large noise (up to 20x COLMAP pose error), and the results can still be improved, although it is not sufficient for high-quality reconstruction.
> > >
> > > - The results in Fig 4 show that the curve could be described by two linear functions. We agree that the curve is related to the scene configuration of the DTU dataset, e.g., the camera pose distribution and scene scales. In a new dataset, this may be different but the trend of a curve should be similar.
> > >
> > > - There is no hard threshold to define the failure case. As we only care about the highly accurate reconstruction, i.e., the case that the optimized pose error is below the initial COLMAP pose error, we could focus more on the first-stage linear curve. It shows that our method can reduce the 1.5xCOLMAP rotation error to about 0.5xCOLMAP rotation error.

---

### Official Review · Reviewer_7s1S · 2023-11-02

**Soundness:** 3 good
**Presentation:** 4 excellent
**Contribution:** 2 fair
**Rating:** 5
**Confidence:** 4

**Summary:**

The authors propose a novel approach in refining the inaccurate camera pose and thereby improving neural surface reconstruction.

The paper presents two key innovations: Pose Residual Field (PoRF) and a robust epipolar geometry loss. PoRF, an implicit pose representation, uses an MLP network to learn pose residuals, considering global information across all frames for enhanced performance. The epipolar geometry loss, used for better supervision, relies on feature correspondences.

The method shows significant improvements in accuracy and efficiency in camera pose refinement. It outperforms existing methods like BARF and SPARF, particularly in datasets like DTU and MobileBrick.

**Strengths:**

- The utilization of global data across all frames by PoRF marks a notable advancement compared to techniques that individually optimize each image. This approach significantly enhances the refinement of camera poses, thereby achieving greater accuracy and efficiency in reconstructions.
- Additionally, the method proves to be highly effective in adjusting poses from various sources, such as COLMAP and ARKit, demonstrating top-tier performance in practical, real-world dataset applications.

**Weaknesses:**

I've noticed that the concept of applying Epipolar Constraints in situations involving inaccurate or absent poses has been previously examined in [1]. It would be interesting to see if there's any discussion or comparison of this aspect in the context of PoRF's approach.

[1] Chen, S., Zhang, Y., Xu, Y., & Zou, B. (2022). Structure-Aware NeRF without Posed Camera via Epipolar Constraint. arXiv preprint arXiv:2210.00183.

**Questions:**

Considering that reference [1] is already available, I would appreciate it if you could provide some insight into how the PoRF Epipolar Geometry loss presents a novel approach compared to the 3D loss mentioned in [1].

---

> ### Author Response · Authors · 2023-11-13
> **Reply to Reviewer 7s1S**
>
> Thank you for your positive comments on the novelty of our proposed PoRF and the evaluation results. In the following, we primarily address concerns related to the difference between our proposed epipolar geometry loss and the 3D-3D loss proposed in [1].
>
> While the primary contribution of our paper lies in the PoRF, our proposed epipolar loss exhibits clear distinctions from previous methods, such as SPARF and [1]. These differences include::
>
> -   [1] computes the 3D-3D loss (Euclidean distance between two 3D points), where the 3D points originate from NeRF.
>
> -   SPARF calculates 3D-2D reprojection errors (projects 3D points to 2D images and computes pixel shifts), with the 3D points also originating from NeRF.
>
> -   Our method computes 2D-2D Sampson errors (the squared distance between a point x and the corresponding epipolar line x'F). Therefore, **our method does not involve NeRF and 3D representation** in computing the epipolar loss.
>
> The 3D-3D loss [1] has been tested in SPARF, and the authors observed that the 3D-3D loss is not as robust as the 3D-2D reprojection loss in the joint optimization of camera poses with NeRF. This is because NeRF may render inaccurate depths. Although the 3D-2D reprojection loss [SPARF] surpasses the 3D-3D loss [1], it still relies on NeRF reconstruction. Consequently, we use 2D-2D Sampson distance, which does not depend on 3D representation. As a result, our method (refer to L3 in Fig. 5), utilizing L_EG without PoRF, consistently outperforms SPARF (refer to Table 1) on the DTU dataset, achieving rotation errors of 0.2 compared to 0.39 degrees.  This illustrates the superiority of our loss design over previous methods.

---

> > ### Comment · Reviewer_7s1S · 2023-11-22
> >
> > Thanks for clarifying my concern. I appreciate your thoroughness and pointing out the differences between losses.

---

### Meta-Review · Area_Chair_2G5A · 2023-12-05

**Metareview:**

This paper aims at jointly optimizing camera pose and geometry of a 3D scene using multi-view images. The key idea include a) a MLP that refines the camera parameters for each view and b) a regularization objective based on Epipolar correspondence. All reviewers recognize that the paper is well written and proposes a simple yet effective method. Weaknesses of the paper include: a) missing comparison with existing work (7s1S), b) time efficiency (7mvY), c) novelty (jgNN) and d) extension to more complex scenes (dcYV). Most of the concerns have been addressed during the rebuttal period.

**Justification For Why Not Higher Score:**

Though this paper proposes a very effective method for joint optimization of camera pose and 3D geometry, it largely builds on existing works. Furthermore, the application scenario and impact are limited to a specific task. Thus with limited contribution, I am not recommending the paper for spotlight or oral representation.

**Justification For Why Not Lower Score:**

The paper proposes a straightforward but effective method for 3D scene reconstruction and camera pose joint optimization. It shows impressive results across multiple datasets. The paper is well written and easy to follow. Thus, I recommend to accept it.

---

### Decision · Program_Chairs · 2024-01-16

Accept (poster)